# Evolution Process of Urban Industrial Land Redevelopment in China: A Perspective of Original Land Users

**Fang He, Yuan Yi**  **and Yuxuan Si *** 

School of Economics and Management, Tongji University, Shanghai 200092, China;
heyoufang@tongji.edu.cn (F.H.); 1810045@tongji.edu.cn (Y.Y.)
*  Correspondence: 2310123@tongji.edu.cn; Tel.: +86-180-1749-6839

**Abstract:** The crucial role of urban industrial land redevelopment in sustainable urban renewal has garnered widespread attention. While some scholars have explored the interest game among stakeholders in industrial land redevelopment, they primarily focus on the government-led model. Moreover, there remains a research gap concerning the impact of government intervention on the redevelopment of industrial land. This article utilizes evolutionary game theory to investigate the interest game between local governments and original land users in the model of urban industrial land redevelopment dominated by original land users. We establish evolutionary game models considering incentives and the combination of incentives and regulations, explore the interest balance strategy, and examine the impact of positive incentives and mandatory regulations on industrial land redevelopment. Furthermore, we employ a numerical simulation to unveil the impact of initial strategies and parameter adjustments on game strategy. The research results are as follows: (1) Under the original land user-led redevelopment model, only two evolutionary stability strategies exist: either the original land users implement industrial land redevelopment with positive responses from local governments, or neither party advances the process. (2) Government intervention is pivotal in facilitating the redevelopment of inefficient industrial land as economic subsidies and punitive measures motivate more participants to adopt proactive strategies. (3) The increase in government support positively correlates with the likelihood of industrial land redevelopment implementation by original land users. (4) The interests and costs of original land users emerge as crucial parameters influencing strategic decisions. This study enriches the understanding of the interests of core participants in industrial land redevelopment and provides valuable insights for sustainable urban renewal.

**Keywords:** sustainable urban renewal; industrial land redevelopment; evolutionary game; government intervention; economic incentive; punitive measure



## 1. Introduction

Sustainable development remains one of the most advocated development concepts worldwide [1,2] and is increasingly being incorporated into national and international development policies [3–5]. The theory of sustainable development, with its emphasis on the harmonious coexistence of economic prosperity, social equity, and environmental preservation, has become increasingly pivotal in the realm of urban renewal [6]. Within this context, sustainable urban renewal assumes considerable significance as a viable approach for augmenting land value and enhancing environmental quality [7]. It serves to rectify urban decline, fulfill socio-economic objectives [8], bolster social networks, and mitigate adverse impacts on residential environments [9]. Given the escalating scarcity of developable land and constricted land resources, industrial land—characterized by expansive acreage, limited developmental yields, and pronounced environmental degradation—emerges as the primary target for urban renewal endeavors [10–13]. The redevelopment of urban industrial land encompasses the revitalization of underutilized or inefficient industrial

areas that fail to meet the demands of urban socio-economic progress [14,15]. Against the backdrop of burgeoning urbanization and urban expansion, the redevelopment of urban industrial land has evinced the potential for sustainable urban renewal, wherein economic advancement is harmonized with social equity and environmental sustainability [16,17]. Developed countries in Europe and America have amassed valuable expertise in legal formulation, financial support, and reuse planning for underutilized industrial land. The redevelopment of industrial land has generated substantial benefits across social, economic, and ecological domains. It is widely regarded by developed countries as indispensable for the sustainable advancement of post-industrial cities and the efficient utilization of land resources [18].

Since the advent of reform and opening up, China has implemented various measures aimed at attracting investment, notably through the provision of low-priced industrial land [14]. This approach has resulted in a notably higher proportion of industrial land within urban construction zones compared to many other nations [19–21]. However, this strategy has brought to light several significant challenges, including issues related to low efficiency and ambiguous property rights concerning industrial land [22,23]. With urbanization progressing unabated, the demand for urban space has reached unprecedented levels, exacerbating the already prominent contradiction between the scarcity of available construction land in numerous large and medium-sized cities [24]. Consequently, the transformation and upgrading of urban industrial land has emerged as a central focus within China's urban renewal [25,26]. Recognizing the urgent need for action, China must swiftly enact comprehensive policies aimed at exploring land utilization methods conducive to socio-economic development, enhancing industrial land efficiency, and ensuring ample space for high-quality urban expansion. Given China's immense economic scale, its strategies and achievements in this domain hold pivotal importance, offering invaluable case studies for global emulation and analysis.

In China, urban land is state-owned. The state transfers land use rights to land users within a certain period of time. Enterprises and individuals can only own land use rights and have ownership of buildings above ground. Considering this, in the context of industrial land redevelopment, this study defines industrial land users as original land users. In many Chinese cities, this redevelopment is predominantly led by the government [27,28]. Specifically, the government pays demolition compensation to the original land users, reclaims land use rights, and subsequently transfers these rights to state-owned enterprises or developers [29]. Remarkably, original land users are not actively engaged in the execution phase of industrial land redevelopment within this model. However, this approach encounters numerous challenges in practice. Firstly, local governments are required to negotiate with original land users regarding compensation for industrial land acquisition, a process often fraught with difficulty in achieving consensus. Additionally, the reluctance of original land users to relinquish their land use rights significantly diminishes their enthusiasm for participating in urban industrial land redevelopment initiatives [30,31].

Shenzhen and Shanghai have pioneered innovative models for industrial land redevelopment, advocating for an approach led by the original land users [19]. In this paradigm, the original land user compensates the government for the difference in land transfer fees resulting from changes in land use and plot ratio, undertaking the redevelopment of urban industrial land under governmental guidance and planning [32]. In contrast to the government-led model, original land users are not required to forfeit their land use rights and can engage in the industrial land redevelopment process. This approach is anticipated to incentivize them to autonomously undertake land redevelopment initiatives. However, the original land users engaged in land redevelopment often experience a reduction in net income compared to maintaining the status quo because of the substantial expenses associated with land transfer fees, employee resettlement, and demolition and reconstruction costs. Consequently, they opt to maintain the current state of affairs.

To facilitate the smooth advancement of urban industrial land redevelopment spearheaded by original land users, the governments of Guangzhou and Shenzhen have devised

corresponding incentive measures, such as allowing original land users to share land appreciation benefits with the government. Moreover, drawing from international experiences in brownfield redevelopment [33], it has become evident that alongside economic incentives, punitive measures serve as an effective mechanism in compelling original land users to undertake redevelopment initiatives [10]. Further investigation is warranted to ascertain whether the intervention of local governments in industrial land redevelopment, specifically through the implementation of reward and punishment measures, will influence the autonomous implementation of land redevelopment by original land users.

The redevelopment of urban industrial land constitutes a multifaceted process entailing a plethora of activities with diverse stakeholders [34]. Central to the successful execution of urban industrial land transformation is the identification of factors shaping stakeholder decision making and the balancing of their interests [35]. While scholars have extensively investigated the influencing factors of urban industrial land redevelopment and the game among core stakeholders within government-led redevelopment models, scant attention has been paid to stakeholder interactions within the model dominated by original land users. Additionally, there exists a notable absence of research concerning the interest game between the government and original land users regarding the implementation of economic subsidies and punitive measures.

This article investigates the interest game between governments and original land users, aiming to provide a scientific basis for policy formulation to achieve a balanced alignment of interests between the two parties. Initially, we develop an evolutionary game model that incorporates economic subsidies. This model facilitates the analysis of the evolutionary trajectory of game behavior, identification of evolutionary stability strategies, and assessment of the impact of parameter variations on both sides of the game. The research reveals that under the redevelopment model dominated by original land users, the evolutionary game between local governments and original land users converges to two final evolutionary stability strategies: implementation and promotion or no implementation and no promotion. Subsequently, this study examines the influence of punitive factors on game participants, constructing an evolutionary game model that incorporates both subsidies and penalties. Finally, the results of the model analysis undergo empirical validation via numerical simulation, exploring the impact of initial strategies and parameter adjustments on the game strategy concerning industrial land redevelopment. Our research underscores the significance of local governments' attitudes toward the redevelopment of urban industrial land in influencing the decision-making processes of core stakeholders. Furthermore, proactive government intervention is pivotal in fostering the autonomous implementation of urban industrial land redevelopment by original land users. Economic subsidies and punitive measures employed by the government increase the probability of both parties implementing industrial land redevelopment. Additionally, the benefits and costs of original land users serve as crucial influencing factors in decision-making processes.

The rest of this paper is organized as follows. Section 2 reviews the literature. Section 3 establishes the evolutionary game model. The numerical simulation and analysis are presented in Section 4. Finally, Section 5 draws the conclusions and shows the policy implications.

## 2. Literature Review

The collaboration among stakeholders is crucial for restoring the economic vitality of industrial land. Therefore, it is necessary to study the behavior of stakeholders in industrial land redevelopment. Additionally, investigating the driving factors of industrial land redevelopment may reveal the factors that predict the completion of redevelopment while assisting stakeholders in the decision-making process, thereby promoting the implementation of industrial land redevelopment [36]. This article provides a literature review from two parts: the demands and interactions of core stakeholders in industrial land redevelopment and the study of factors affecting land redevelopment.

## 2.1. Research on Stakeholders of Urban Industrial Land Redevelopment

Limited research exists on stakeholders in urban redevelopment within the current literature, primarily emphasizing the major stakeholders and their interest game behaviors. Some scholars have studied the ideas and demands of core stakeholders such as local governments, consultants, original land owners, original land users, new developers and the public in industrial land redevelopment projects and found that different stakeholder groups have great differences in economic, social and environmental expectations [29,37].

The redevelopment of urban industrial land in China is a complex process involving numerous activities carried out by many stakeholders. Scholars have adopted game theory methods to explore the dynamic interaction among stakeholders, which is described as a bounded rational decision-making problem characterized by value maximization [34]. Some scholars have initiated their inquiry from the fundamental mechanism of spatial games, subsequently constructing an ideal game model grounded on value equilibrium to scrutinize the governance models of Changzhou and Shenzhen. Through this analysis, they endeavor to explore the direction of institutional innovation concerning the renewal of urban industrial parks [38]. Scholars posit that the redevelopment of industrial land in Chinese cities can be theoretically elucidated as a multiple game involving core stakeholders [34]. The relevant research adopts a perspective centered on multi-party interest games, probing into the contradictions and challenges inherent in the interest game among the government, market entities, and original property owners during industrial land renewal [39]. Additionally, there is research dedicated to examining the game strategy between the original and new property rights holders of inefficient industrial land and local governments. This research suggests generating incremental benefits by adjusting planning and construction indicators and land management methods, thus fostering a balance of interests among different property rights holders [40]. Moreover, scholars have developed three game theory models to analyze the game processes of key stakeholders in three distinct types of redevelopment projects [27,31,32]. Notably, the aforementioned research primarily concentrates on the interest game of stakeholders within the government-led model.

## 2.2. Research on Factors Affecting the Redevelopment of Industrial Land

Extensive scholarly research has been devoted to exploring the driving and hindering factors influencing industrial land redevelopment. It has been emphasized that the strategic planning of economic development zones significantly impacts the expansion of industrial land, while both land prices and population density wield profound influence over its redevelopment [41]. Particularly, the escalating expectations associated with land prices serve as pivotal determinants in the urban industrial land redevelopment process [21]. Furthermore, pivotal driving factors include pollution mitigation, the augmentation of employment opportunities, and the implementation of cultural development strategies [42,43]. Conversely, obstacles to industrial land redevelopment encompass the uncertainty surrounding redevelopment policies, deficient trust between local governments and original land users, the prolonged reliance of original land users on land transfer income, and the high transaction costs involved in reaching a consensus [29,30]. Moreover, the substantial cost of implementation and market demand uncertainty pose further challenges to industrial land redevelopment initiatives [42]. Notably, the study reveals that the correlation between the absence of legal land rights and redevelopment outcomes lacks significance [44].

## 2.3. Research Gap

At present, while a small portion of the literature has investigated the game of interests among stakeholders, it predominantly centers on the government-led model. In this model, original land users are required to forfeit their land use rights and are not engaged in the process of industrial land redevelopment, resulting in a lack of enthusiastic response. Guided by local governments, the self-development model of industrial land by original land users is anticipated to enhance their enthusiasm. Significantly, scholars have yet to investigate this aspect from the perspective of original land users as the main players.

In addition, scholars have paid attention to the influencing factors of urban industrial land redevelopment. However, there is a gap in the research regarding the impact of government interventions such as economic incentives and punitive measures on the redevelopment of industrial land.

Exploring the interest game relationships among core participants and the role of the government in promoting industrial land redevelopment in the context of the autonomous implementation of urban industrial land redevelopment by original land users is beneficial for enriching land planning and management theory, as well as for promoting the process of industrial land redevelopment. This exploration holds both theoretical and practical significance.

### 3. Evolutionary Game Model

Evolutionary game theory is a theoretical framework that combines game theory with evolutionary biology to study the evolution process of individual strategies in natural selection and population dynamics [45]. The core idea of the theory is that an individual's behavioral strategies in a group evolve over time to adapt to the environment and interactions within the group [46]. Different from the traditional game theory, evolutionary game theory holds that human rationality is limited and the complete information conditions are unnecessary [47]. The theory has been widely applied in various fields to analyze the strategic choices and behavioral evolution of individuals and groups in complex environments.

The redevelopment of urban industrial land holds significant potential for fostering government fiscal revenue growth, local economic prosperity, and environmental enhancement [42]. Consequently, there exists a strong governmental impetus to drive forward such redevelopment initiatives. However, it is imperative to acknowledge that original land users invariably prioritize maximizing their economic gains in any situation [37]. In the process of urban industrial land redevelopment, the decision-making behaviors of the two stakeholder groups, the government and the original land users, influence each other, warranting exploration through the lens of evolutionary game theory [27,34]. Therefore, we designate both the government and original land users as participants in the evolutionary game model.

Based on the research of scholars on the influencing factors of industrial land redevelopment, this study analyzes the game behavior of stakeholders in the decision-making process of urban industrial land redevelopment under different scenarios, exploring the final equilibrium strategy and influencing factors. Firstly, we construct an evolutionary game model considering subsidies, analyzing the evolutionary path of game behavior, evolutionary stability strategies, and the impact of different parameter changes on evolutionary stability strategies. Then, we add the penalty factor to the basic model and construct an evolutionary game model combining subsidies and penalties, aiming to explore the interest balance strategy and examine the impact of incentive policies and mandatory regulations on industrial land redevelopment.

#### 3.1. Model Assumptions

Before constructing the evolutionary game model, we establish the following four assumptions to reflect the actual situation of urban industrial land redevelopment.

**Assumption 1.** *The government and original land users independently adopt behavioral strategies and dynamically adjust their strategies. The primary factor influencing stakeholder decision making is personal interests [34]. Original land users are focused on maximizing their own economic gains, while the government places priority on serving social interests.*

**Assumption 2.** *To analyze the game behavior of both parties, we define their selection strategies based on practical considerations. The government and the original land users have two strategic options. For the government, one strategy is to promote the redevelopment of urban industrial land by offering economic subsidies to original land users, derived from a certain proportion of land*

*appreciation benefits. Another strategy is to refrain from taking any action. For original land users, one strategy is to opt for implementing urban industrial land redevelopment, while the alternative is to reject such redevelopment.*

**Assumption 3.** *If governments choose "promote", they need to pay additional costs for facilitating the redevelopment of urban industrial land, including investments in human and material resources as well as economic subsidies. However, when original land users choose to implement redevelopment, the government stands to gain significant social benefits. These additional benefits may include fostering a positive government image, enhancing satisfaction among the original land user group, and improving government performance [34]. In the long run, the additional benefits accruable to the government are expected to outweigh the extra costs incurred in promoting the redevelopment of urban industrial land.*

**Assumption 4.** *If original land users choose "implement", they can expect to receive higher benefits compared to maintaining the status quo, despite incurring costs such as reconstruction and land transfer fees.*

### 3.2. Establishment of Economic Incentive Model

3.2.1. Model Establishment

If governments choose "not promote", and original land users are willing to implement land redevelopment, industrial land redevelopment can still be accomplished. The expected benefits that the original land users can obtain are denoted as $R'$, encompassing the benefits derived from land redevelopment. They are required to cover the costs of demolishing and rebuilding buildings, corporate income tax, and the difference in land transfer fees resulting from changes in plot ratio or land use. These costs are defined as $C_E + \lambda(R' - C_E - C_L) + C_L$. And, the governments benefit from land transfer fees, income tax revenue, and environmental benefits, defined as $C_L + \lambda(R' - C_E - C_L) + R_{G1}$, without paying any costs.

If the governments choose not to promote, the original land users are unwilling to implement land redevelopment, industrial land renewal and transformation cannot be completed. The profit of original land users is the income obtained from maintaining the status quo minus the cost of income tax, defined as $(1 - \lambda)R$, while the income obtained by governments is $\lambda R$.

If the governments choose the "promote" strategy and the original land users agree to implement industrial land redevelopment, in addition to land transfer fee income, income tax income, and environmental benefits, the government can also receive social benefits, defined as $C_L + \lambda(R' - C_E - C_L) + R_{G1} + R_{G2}$. However, governments need to cover economic incentive costs, including subsidies and communication expenses, as well as the costs associated with formulating policies and promoting implementation, which can be defined as $C_{G1} + C_{G2}$. For the original land users, they can receive benefits from land redevelopment and subsidies provided by the government, namely $R' + \beta C_L$, with costs including demolition and reconstruction costs, corporate income tax, and the difference between land transfer fees, defined as $C_E + \lambda(R' - C_E - C_L) + C_L$.

If the governments choose the strategy of "promote" and the original land users refuse to implement land redevelopment, land redevelopment cannot be carried out. In such a scenario, the benefits of the original land users can be recognized as $(1 - \lambda)R$. The expected revenue of the government is the tax revenue from maintaining the status quo of industrial land, which is denoted as $\lambda R$. However, governments need to bear the cost of formulating and implementing policies, defined as $C_{G2}$.

The specific parameter settings are outlined in Table 1, based on the problem description and assumptions.

Based on specified assumptions and parameter configurations, we establish a game payoff matrix delineating interactions between the original land users and the government, as presented in Table 2, wherein all parameters assume non-negative values. Within each cell, the first row denotes the income of the original land users, while the second row

represents the income of the government. The first cell represents the two benefits if governments choose "promote" and the original land users choose "implement". The return of original land users is $(1-\lambda)(R'-C_E-C_L)+\beta C_L$, and the governments receive $\lambda(R'-C_E-C_L)+C_L+R_{G1}+R_{G2}-C_{G1}-C_{G2}$.

**Table 1.** Specific parameter settings.

| Parameter | Description |
|---|---|
| $\lambda$ | Corporate income tax rate, $1 > \lambda > 0$ |
| $R'$ | Income of original land users if they implement land redevelopment, $R' > R > 0$ |
| $R$ | Income of original land users if they do not implement land redevelopment, $R > 0$ |
| $C_E$ | Cost of demolition and reconstruction for original land users if they implement land redevelopment, $C_E > 0$ |
| $C_L$ | Cost of the difference in land transfer fees paid by original land users due to changes in plot ratio or land use if they implement land redevelopment, $C_L > 0$ |
| $\beta$ | Proportion of economic subsidies if the governments choose "promote" and the original land users choose "implement", $1 \geq \beta \geq 0$ |
| $R_{G1}$ | Environmental benefits of governments if governments choose "not promote" and the original land users choose "implement", $R_{G1} > 0$ |
| $R_{G2}$ | Social benefits of governments if governments choose "promote" and the original land users choose "implement", $R_{G2} > 0$ |
| $C_{G1}$ | Cost of economic subsidies and communication paid by governments if governments choose "promote" and the original land users choose "implement", $C_{G1} > 0$ |
| $C_{G2}$ | Cost of formulating and promoting policies paid by governments if governments choose "promote", $C_{G2} > 0$ |

**Table 2.** Game payoff matrix between the original land user and governments.

| Original Land Users | Governments | |
|---|---|---|
| | **Promote (*y*)** | **Not Promote (1−*y*)** |
| Implement (*x*) | $(1-\lambda)(R'-C_E-C_L)+\beta C_L$ | $(1-\lambda)(R'-C_E-C_L)$ |
| | $\lambda(R'-C_E-C_L)+C_L+R_{G1}+R_{G2}-C_{G1}-C_{G2}$ | $\lambda(R'-C_E-C_L)+C_L+R_{G1}$ |
| Not Implement 1 − (*x*) | $(1-\lambda)R$ | $(1-\lambda)R$ |
| | $\lambda R - C_{G2}$ | $\lambda R$ |

At the onset of evolution, the proportion of original land users opting for "implementation" is denoted as $x$ ($0 \leq x \leq 1$), whereas the proportion choosing "not implement" is represented as $1-x$. Correspondingly, the proportion of governments electing "promote" is labeled as $y$ ($0 \leq y \leq 1$), and the proportion opting for "not promote" is indicated as $1-y$.

This paper defines the anticipated returns of original land users for engaging in urban industrial land redevelopment and abstaining from land redevelopment as $W_L^1$ and $W_L^2$, respectively. The mean expected return of original land users is symbolized as $\overline{W_L}$. The equations are as follows:

$$W_L^1 = y\big[(1-\lambda)\big(R'-C_E-C_L\big)+\beta C_L\big] + (1-y)\big[(1-\lambda)\big(R'-C_E-C_L\big)\big] \quad (1)$$

$$W_L^2 = y[(1-\lambda)R] + (1-y)[(1-\lambda)R] \quad (2)$$

$$\overline{W_L} = xW_L^1 + (1-x)W_L^2 \quad (3)$$

Following Equations (1)–(3), the replication dynamic equation for the selection strategy of the original land users is formulated in Equation (4). Here, *t* represents time, and $dx/dt$ signifies the rate of change over time in the proportion of original land users opting to implement industrial land redevelopment.

$$F_{(x)} = dx/dt = x(W_L^1 - \overline{W_L}) = x(1-x)\big[(1-\lambda)\big(R'-C_E-C_L-R\big)+y\beta C_L\big] \quad (4)$$

This assumes that the anticipated returns on promoting and abstaining the redevelopment of industrial land by the government are $W_G^1$ and $W_G^2$, respectively, with the average expected return of the entire government group set as $\overline{W_G}$, as depicted in Formulas (5)–(7).

$$W_G^1 = x\big[\lambda\big(R' - C_E - C_L\big) + C_L + R_{G1} + R_{G2} - C_{G1} - C_{G2}\big] + (1 - x)(\lambda R - C_{G2}) \quad (5)$$

$$W_G^2 = x\big[\lambda\big(R' - C_E - C_L\big) + C_L + R_{G1}\big] + (1 - x)\lambda R \quad (6)$$

$$\overline{W_G} = yW_G^1 + (1 - y)W_G^2 \quad (7)$$

The replication dynamic equation for government selection strategies is presented in Equation (8) where $dy/dt$ represents the rate of change over time in the proportion of governments opting to promote industrial land redevelopment.

$$F_{(y)} = dy/dt = y(W_G^1 - \overline{W_G}) = y(1 - y)[x(R_{G2} - C_{G1}) - C_{G2}] \quad (8)$$

The model reaches a stable state and ceases evolving when the dynamic replication equation equals 0 [48]. By setting $F_{(x)} = 0$ and $F_{(y)} = 0$, we derive $E_1(0,0)$, $E_2(0,1)$, $E_3(1,0)$, $E_4(1,1)$, and $E_5(A, B)$ as the five equilibrium points for the dynamic game matrix.

$$A = C_{G2}/(R_{G2} - C_{G1}) \quad (9)$$

$$B = (1 - \lambda)(R + C_E + C_L - R')/\beta C_L \quad (10)$$

### 3.2.2. Model Analysis

The five equilibrium points derived from the replication dynamic equations represented by Formulas (4) and (8) are not ascertainable as the evolutionary stability strategy within this system. To ascertain this, the methodology advocated by Friedman is employed, utilizing Jacobian matrix stability analysis to evaluate whether these points represent evolutionary equilibrium states [49]. The Jacobian matrix serves as a precise linear approximation of a differentiable equation at a specific point. Through the analysis of the Jacobian matrix, we can ascertain whether the equilibrium points indeed constitute evolutionary stable strategies [50].

$$\mathbf{J} = \begin{bmatrix} \partial F_{(y)}/\partial y & \partial F_{(y)}/\partial x \\ \partial F_{(x)}/\partial y & \partial F_{(x)}/\partial x \end{bmatrix} = \begin{bmatrix} a_{11} & a_{12} \\ a_{21} & a_{22} \end{bmatrix} \quad (11)$$

In Equation (11), $a_{11} = (1 - 2y)[x(R_{G2} - C_{G1}) - C_{G2}]$, $a_{12} = y(1 - y)(R_{G2} - C_{G1})$, $a_{21} = x(1 - x)\beta C_L$, and $a_{22} = (1 - 2x)[(1 - \lambda)(R' - C_E - C_L - R) + y\beta C_L]$.

Equations (12) and (13) illustrate the determinant equation and trace of the Jacobian matrix.

$$\det(\mathbf{J}) = a_{11}a_{22} - a_{12}a_{21} \quad (12)$$

$$\text{tr}(\mathbf{J}) = a_{11} + a_{22} \quad (13)$$

An equilibrium point qualifies as an evolutionarily stable strategy if $\det(\mathbf{J}) > 0$ and $\text{tr}(\mathbf{J}) < 0$. If $\det(\mathbf{J}) > 0$ and $\text{tr}(\mathbf{J}) > 0$, the equilibrium point is deemed unstable. When $\det(\mathbf{J}) < 0$, the equilibrium point is the saddle point. The five equilibrium points are substituted into the determinant equation and trace, with the outcomes summarized in Table 3.

The signs of $\det(\mathbf{J})$ and $\text{tr}(\mathbf{J})$ are determined by four parts: $C_{G2}$, $(1 - \lambda)(R' - C_E - C_L - R)$, $(1 - \lambda)(R' - C_E - C_L - R) + \beta C_L$, and $R_{G2} - C_{G1} - C_{G2}$. We confirm that $C_{G2} > 0$ and $R_{G2} - C_{G1} - C_{G2} > 0$. Therefore, the evolutionary stability strategy depends on the signs of the other two parts.

**Scenario 1.** $0 < (1 - \lambda)(R' - C_E - C_L - R) < (1 - \lambda)(R' - C_E - C_L - R) + \beta C_L$

**Table 3.** Determinant equation and trace of five equilibrium points.

| Equilibrium | det(J) | tr(J) |
|---|---|---|
| $E_1(0,0)$ | $-C_{G2} \times (1-\lambda)(R' - C_E - C_L - R)$ | $-C_{G2} + (1-\lambda)(R' - C_E - C_L - R)$ |
| $E_2(0,1)$ | $C_{G2} \times [(1-\lambda)(R' - C_E - C_L - R) + \beta C_L]$ | $C_{G2} + (1-\lambda)(R' - C_E - C_L - R) + \beta C_L$ |
| $E_3(1,0)$ | $-(R_{G2} - C_{G1} - C_{G2}) \times$ $(1-\lambda)(R' - C_E - C_L - R)$ | $(R_{G2} - C_{G1} - C_{G2}) -$ $(1-\lambda)(R' - C_E - C_L - R)$ |
| $E_4(1,1)$ | $(R_{G2} - C_{G1} - C_{G2}) \times$ $[(1-\lambda)(R' - C_E - C_L - R) + \beta C_L]$ | $-(R_{G2} - C_{G1} - C_{G2}) -$ $[(1-\lambda)(R' - C_E - C_L - R) + \beta C_L]$ |
| $E_5(A,B)$ | 0 | 0 |

In this scenario, compared to refraining from implementing industrial land redevelopment, original land users who undertake redevelopment experience higher returns, signaling the success of their transformation projects. Economic subsidies from the government further enhance their earnings potential. The results of local stability analysis for four equilibrium points are detailed in Table 4, with the non-existence of equilibrium point $E_5$ due to $B = (1-\lambda)(R + C_E + C_L - R')/\beta C_L < 0$.

**Table 4.** Local stability analysis in Scenario 1.

| Equilibrium | det(J) | tr(J) | Result |
|---|---|---|---|
| $E_1(0,0)$ | Negative | Uncertain | Saddle |
| $E_2(0,1)$ | Positive | Positive | Unstable |
| $E_3(1,0)$ | Negative | Uncertain | Saddle |
| $E_4(1,1)$ | Positive | Negative | Stable |

The dynamic evolution path of the equilibrium points in Scenario 1 is as follows: the points start from $E_2$, pass through $E_1$ and $E_3$, and finally converge to $E_4$, which stands as the sole evolutionarily stable strategy in Scenario 1. The ultimate strategic choice is for the government to promote the redevelopment of industrial land and the original land users agree to implement it.

**Scenario 2.** $(1-\lambda)(R' - C_E - C_L - R) < 0 < (1-\lambda)(R' - C_E - C_L - R) + \beta C_L$

For original land users, without government subsidies, the benefits from land redevelopment are inferior to maintaining the status quo. However, with subsidies, the redevelopment benefits surpass those of the status quo. It can be obtained that $(1-\lambda)(R + C_E + C_L - R') < \beta C_L$. $E_5$ is a possible equilibrium point because $A$ and $B$ are both in the [0, 1] interval. Table 5 outlines the outcomes of local stability analysis for five equilibrium points.

**Table 5.** Local stability analysis in Scenario 2.

| Equilibrium | det(J) | tr(J) | Result |
|---|---|---|---|
| $E_1(0,0)$ | Positive | Negative | Stable |
| $E_2(0,1)$ | Positive | Positive | Unstable |
| $E_3(1,0)$ | Positive | Positive | Unstable |
| $E_4(1,1)$ | Positive | Negative | Stable |
| $E_5(A,B)$ | Negative | 0 | Saddle |

Figure 1 portrays the dynamic evolution path of the equilibrium points in Scenario 2, with arrows representing the direction of points movement. The points start from $E_2$ and $E_3$, pass through $E_5$, and eventually converge to $E_2$ and $E_4$. This delineates the existence of two evolutionarily stable strategies: (not implement, not promote) and (implement, promote).

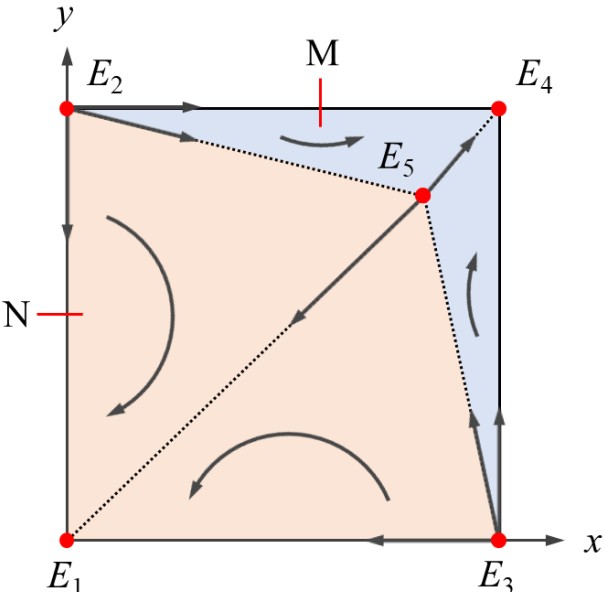

**Figure 1.** The dynamic evolution path of equilibrium points in Scenario 2.

**Scenario 3.** $(1-\lambda)(R'-C_E-C_L-R) < (1-\lambda)(R'-C_E-C_L-R)+\beta C_L < 0$

For original land users, even with government subsidies, the benefits from land redevelopment remain inferior to maintaining the status quo. It can be concluded that $(1-\lambda)(R+C_E+C_L-R') > \beta C_L$, precluding the equilibrium point $E_5$. Table 6 presents the results of local stability analysis for four equilibrium points.

**Table 6.** Local stability analysis in Scenario 3.

| Equilibrium | det(J) | tr(J) | Result |
|---|---|---|---|
| $E_1(0,0)$ | Positive | Negative | Stable |
| $E_2(0,1)$ | Negative | Uncertain | Saddle |
| $E_3(1,0)$ | Positive | Positive | Unstable |
| $E_4(1,1)$ | Negative | Uncertain | Saddle |

The dynamic evolution path of the equilibrium points in Scenario 3 is as follows: the points start from $E_3$, pass through $E_4$ and $E_2$, and finally converge to $E_1$, which is the only evolutionarily stable strategy in Scenario 3. The ultimate strategic choice is for the government not to promote the redevelopment of industrial land and for original land users not to implement land redevelopment. This implies that economic subsidies have no incentivizing effect on promoting urban industrial land redevelopment, which is not an ideal situation.

Scenario 2 delineates a situation where some governments promote redevelopment, while some original land users implement it, with the remainder making opposite decisions. Figure 1 presents two distinct strategies, with the results determined by the areas of M and N. The likelihood of (implement, promote) and (not implement, not promote) is equivalent when $S_M = S_N$. When $S_M > S_N$, a greater number of players opt for (implement, promote), and this result is expected because the redevelopment of industrial land contributes to sustainable urban development. Conversely, more participants choose (not implement, not promote) when $S_M < S_N$, indicating that government incentives are ineffective.

3.2.3. Impacts of Parameters Change

The area of M is determined by A and B, and the area formula is as follows:

$$S_M = \left[2 - (1-\lambda)(R+C_E+C_L-R')/\beta C_L - C_{G2}/(R_{G2}-C_{G1})\right]/2 \qquad (14)$$

Seven parameters influencing the evolutionary stability strategy are identified in Formula (13). The impact of specific parameter alterations is delineated in Table 7, leading to key conclusions.

**Table 7.** Impact of parameter changes.

| Parameter Changes | $R$ | $R'$ | $C_E$ | $\beta$ | $C_{G1}$ | $R_{G2}$ | $C_{G2}$ |
|---|---|---|---|---|---|---|---|
| | ↑ | ↑ | ↑ | ↑ | ↑ | ↑ | ↑ |
| $S_M$ | ↓ | ↑ | ↓ | ↑ | ↓ | ↑ | ↓ |
| $S_N$ | ↑ | ↓ | ↑ | ↓ | ↑ | ↓ | ↑ |

Holding other parameters constant, an increase in the benefits derived from maintaining the status quo ($R$), demolition and reconstruction costs ($C_E$), government economic incentive costs ($C_{G1}$), and promotion policy costs ($C_{G2}$) correlates with a decrease in $S_M$, elevating the likelihood of the model converging towards the equilibrium point $E_1(0,0)$. Consequently, more governments and original land users will choose (implement, promote) if the benefits of maintaining the status quo, demolition and reconstruction costs, economic incentives, and the cost of implementing policies are reduced. If the benefits of redevelopment ($R'$), the proportion of value-added benefits from shared land for original land users ($\beta$), and the social benefits obtained by the governments ($R_{G2}$) are increased, more governments and original land users will choose (implement, promote). Conversely, their strategy is (not implement, not promote).

Through the analysis presented above, it is evident that the costs and benefits incurred by original land users in implementing industrial land redevelopment, along with the costs and social benefits associated with the government promotion of this activity, exert a significant influence on system stability strategies. Moreover, under government policy intervention, local governments can appropriately increase economic subsidies for land redevelopment undertaken by original land users, which helps to increase their enthusiasm for participating in land redevelopment, thereby fostering the implementation of inefficient industrial land redevelopment. Incentive policies play a pivotal role in optimizing land resource utilization and promoting sustainable urban development. Consequently, enhancing the mechanism for sharing land appreciation benefits between the government and original land users holds considerable practical significance.

Considering the actual situation, the income tax rate ($\lambda$) is determined by the central government, and local governments have no right to adjust it. Secondly, the difference in land transfer fees paid due to changes in plot ratio or land use ($C_L$) is defined as the objective value of development rights, which is a fixed and unchanging parameter. Therefore, these parameters exert no influence on the outcome.

*3.3. Establishment of the Model Combining Government Incentives and Punishments*

The analysis indicates a positive incentivizing effect of economic subsidies on original land users for implementing land redevelopment. Furthermore, the punitive mechanism is deemed effective in promoting industrial land redevelopment [10]. The combined application of subsidies and punitive measures yields a superior outcome in fostering urban industrial land redevelopment.

For original land users, the implementation of the punitive mechanism implies a decrease in profits if they opt to maintain the status quo while the government actively promotes land redevelopment. Local governments augment profits by raising water and electricity prices or imposing environmental protection fees on original land users. It is postulated in this article that the defiance of government promotion policies by original land users results in immediate punishment [51], with the fine denoted as $F$. The payoff matrix under the amalgamation of subsidies and punitive measures is depicted in Table 8. Within each cell, the first row denotes the income of the original land users, while the second row represents the income of the government.

**Table 8.** Payoff matrix under the combination of subsidies and punishments.

| Original Land Users | Governments | |
| --- | --- | --- |
| | **Promote (*y*)** | **Not Promote (1−*y*)** |
| Implement (*x*) | $(1-\lambda)(R'-C_E-C_L)+\beta C_L$ | $(1-\lambda)(R'-C_E-C_L)$ |
| | $\lambda(R'-C_E-C_L)+C_L+R_{G1}+R_{G2}-C_{G1}-C_{G2}$ | $\lambda(R'-C_E-C_L)+C_L+R_{G1}$ |
| Not Implement $1-(x)$ | $(1-\lambda)R-F$ | $(1-\lambda)R$ |
| | $\lambda R-C_{G2}+F$ | $\lambda R$ |

The outcomes of the evolutionary game model, integrating both subsidy and punishment mechanisms, closely mirror those of the model focusing solely on subsidy mechanisms. The five equilibrium points within the combined subsidy and punishment framework are as follows: $E_1(0,0)$, $E_2(0,1)$, $E_3(1,0)$, $E_4(1,1)$, and $E_5'(A', B')$, where $A' = (C_{G2}-F)/(R_{G2}-C_{G1}-F)$ and $B' = (1-\lambda)(R+C_E+C_L-R')/(\beta C_L+F)$.

A comparative analysis is conducted between the equilibrium points $E_5$ and $E_5'$. Regarding the relationship between $A'$ and $A$, the subtractive value being less than zero signifies $A' < A$. Similarly, it is evident that $B' = (1-\lambda)(R+C_E+C_L-R')/(\beta C_L+F)$ is inferior to $B = (1-\lambda)(R+C_E+C_L-R')/\beta C_L$. Notably, the position of $E_5'$ shifts, with the point relocating towards the left and downward in comparison to $E_5$. This observation indicates an expansion in the area of M, signifying an augmented likelihood of the system converging towards $E_4(1,1)$. Consequently, a greater number of participants are inclined to opt for the strategy of (implement, promote).

The area of M is determined by $A'$ and $B'$, and the area formula is as follows:

$$S_M = \left[2 - (1-\lambda)(R+C_E+C_L-R')/(\beta C_L+F) - (C_{G2}-F)/(R_{G2}-C_{G1}-F)\right]/2 \qquad (15)$$

The area of M is influenced by eight parameters and escalates with the increase in parameter *F*, while the effects of other parameter adjustments remain consistent with the model outcomes derived from subsidy scenarios. The findings suggest that governmental punitive measures targeting original land users who decline to undertake industrial land redevelopment prove effective in promoting its implementation. This underscores the significant role of appropriate government intervention in enhancing the efficiency of industrial land redevelopment. Both punitive and incentive measures demonstrate comparable effectiveness in this regard.

## 4. Numerical Simulation and Analysis

This paper empirically validates the conclusions drawn from the game model incorporating subsidy and punishment mechanisms through specific numerical simulations, employing MATLAB R2018b to examine the influence of initial strategies on game outcomes (Supplementary Materials). Furthermore, by analyzing the impact of the initial strategy and parameter adjustments on the game results, this study explores the key role of the government in promoting the process of industrial land redevelopment, which holds significant practical implications.

The existence of a large amount of inefficient industrial land is an obstacle for Shanghai to achieve sustainable development in the context of post industrialization. Putuo District, located in the central area of Shanghai, has a significant quantity of inefficient and abandoned industrial land, which seriously restricts the development of the area. Hence, this study selected Putuo District in Shanghai as the primary source of simulation values.

Our evolutionary game model comprises 11 parameters, with initial values set as presented in Table 9. These values, including land redevelopment benefits ($R'$), income from maintaining the status quo ($R$), demolition and reconstruction costs ($C_E$), and differences in land transfer fees due to changes in plot ratio or land use ($C_L$), are derived from the actual context of the Changzheng Industrial Park redevelopment project in Putuo District, Shanghai. Adhering to the implementation regulations of the Enterprise Income

Tax Law of the People's Republic of China, the enterprise income tax rate ($\lambda$) is set at 0.25 [52]. Additionally, the initial values of other parameters, including subsidy ratios ($\beta$), environmental benefits ($R_{G1}$), social benefits ($R_{G2}$), subsidy and communication costs ($C_{G1}$), policy development costs ($C_{G2}$), and fines ($F$), are set by three experts with senior titles and more than 10 years of experience in urban industrial land redevelopment.

**Table 9.** Initial value setting of the parameters.

| Parameter | $R'$ | $R$ | $C_E$ | $C_L$ | $\lambda$ | $\beta$ | $R_{G1}$ | $R_{G2}$ | $C_{G1}$ | $C_{G2}$ | $F$ |
|---|---|---|---|---|---|---|---|---|---|---|---|
| Value | 1100 | 585 | 250 | 320 | 0.25 | 0.2 | 40 | 160 | 85 | 25 | 10 |

### 4.1. Simulation of the Evolutionary Game Model

Given that Scenario 2 presents two evolutionarily stable strategies, differing from Scenarios 1 and 3, we simulate the game outcomes of Scenario 2 within the subsidy game model, depicted in Figure 2. The simulation results indicate stable points at (0, 0) and (1, 1), corresponding to the evolutionary stability strategies of (not implement, not promote) and (implement, promote), which verifies the correctness of the model analysis in Section 3.2. This suggests that, solely relying on market mechanisms, original land users lack the intrinsic motivation to independently engage in urban industrial land redevelopment. Despite the higher benefits from implementing such redevelopment compared to maintaining the status quo, original land users face substantial costs such as demolition and resettlement fees and land transfer fees. In most instances, the marginal difference between the benefits and costs of redevelopment is smaller than the benefits of maintaining the status quo, elucidating the fundamental reluctance of original land users to actively pursue redevelopment. These findings underscore the importance for governments to implement incentive policies, such as providing economic subsidies, to foster urban industrial land redevelopment.

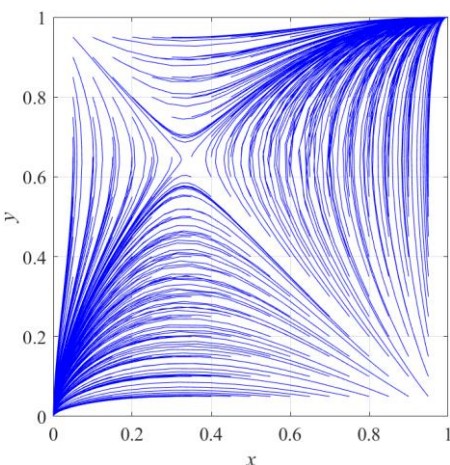

**Figure 2.** Numerical simulation of Scenario 2 in the game model considering subsidies.

In Section 3.3, we observed that the shift in equilibrium point $E_5$ heightens the likelihood of game players selecting (implement, promote) strategies. To empirically validate this observation, we simulate Scenario 2 within the evolutionary game model considering both subsidies and penalties, as depicted in Figure 3. In the numerical simulation diagram, the hollow segment of the line converging to (0, 0) and (1, 1) represents the fifth equilibrium point. From the initial parameter values, we compute $E_5$ as (0.33, 0.64) and $E_5'$ as (0.23, 0.56). Notably, the expansion model exhibits a greater participation rate in implementing and promoting urban industrial land redevelopment, indicating the efficacy of punitive measures in augmenting the likelihood of original land users opting for redevelopment. These findings corroborate the validity of the research conclusion regarding the model of combining incentives and punishments as outlined in Section 3.3.

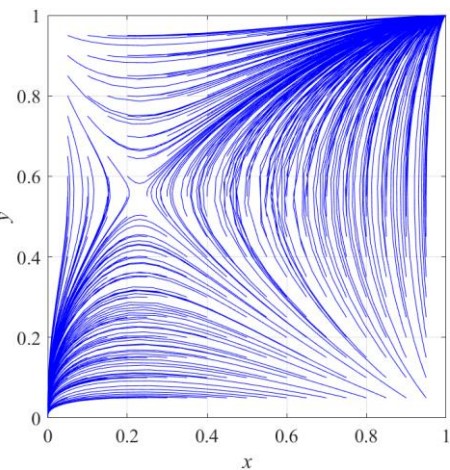

**Figure 3.** Numerical simulation of Scenario 2 in the game model considering subsidies and penalties.

*4.2. Impact of Different Initial Strategies*

Given the presence of two evolutionarily stable strategies in Scenario 2, we explore whether different initial strategies of game players influence the model outcomes [53,54]. Specifically, we vary the initial proportion of local governments selecting the "promote land redevelopment" strategy ($y_0$) in Scenario 2 while keeping other parameters constant. With $y_0$ fixed at 0.4 or 0.7, $x_0$ assumes values ranging from 0.1 to 0.9, as illustrated in Figure 4.

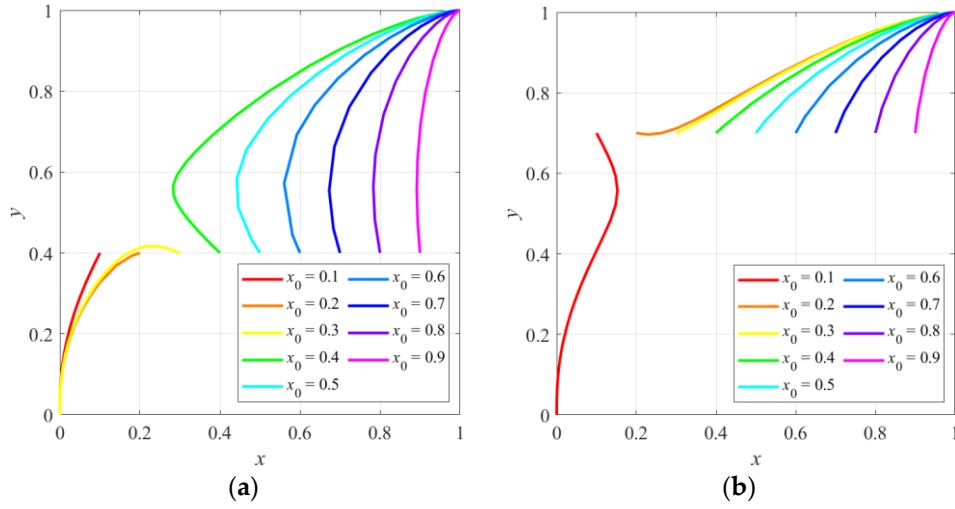

**Figure 4.** Diagram on the impact of different initial government strategies. (**a**) $y_0 = 0.4$. (**b**) $y_0 = 0.7$.

When $y_0$ is fixed at 0.4, the initial point with a horizontal axis $x_0 \geq 0.4$ converges to (1, 1), while the initial point with a horizontal axis $x_0 < 0.4$ converges to (0, 0). If the initial strategy $y_0$ is fixed at 0.7, the initial point with a horizontal axis $x_0 \geq 0.2$ converges to (1, 1), while the rest converges to (0, 0). The conclusion drawn is that as the initial proportion of local governments opting for the strategy of "promoting industrial land redevelopment" increases, the likelihood of the system stabilizing at the (implement, promote) strategy also increases. Based on the analysis above, the more local governments promote the redevelopment of urban industrial land, the more likely the original land users are to agree to implement it. Therefore, the government should adopt a positive and proactive attitude towards the independent implementation of industrial land redevelopment by original land users.

### 4.3. Impact of Parameter Changes on Strategy Selection

Parameter changes significantly influence the outcomes of evolutionary game models [51,54], potentially altering the evolutionary stability strategy in Scenario 2. To investigate the influence of factors on the game strategy of industrial land redevelopment, we set the initial strategy as $y_0 = 0.5$ and $x_0 = 0.5$, and analyze the effects of the original land users' benefits ($R'$ and $R$), demolition and reconstruction costs ($C_E$), economic subsidies ($\beta$), and penalties ($F$) on strategy selection.

#### 4.3.1. Expected Returns and Costs of Original Land Users

Keeping other parameters constant, we vary $R'$ from 1000 to 1200 and observe its impact on $x$ and $y$ in Figure 5. As $R'$ increases, $x$ and $y$ transition from 0 to 1, indicating a shift in strategy selection from (0, 0) to (1, 1). This suggests that higher returns for original land users lead to a greater inclination of the government to promote urban old industrial land redevelopment, with original land users more willing to implement the redevelopment. Figure 6 illustrates the impact of changes in $R$ values on $x$ and $y$, showing an opposite trend to that of $R'$. As $R$ increases, $x$ and $y$ change from 1 to 0, shifting the selection strategy from (1, 1) to (0, 0). This implies that reduced benefits from maintaining the current land state prompt original land users to implement industrial land redevelopment, while the government chooses to actively promote it.

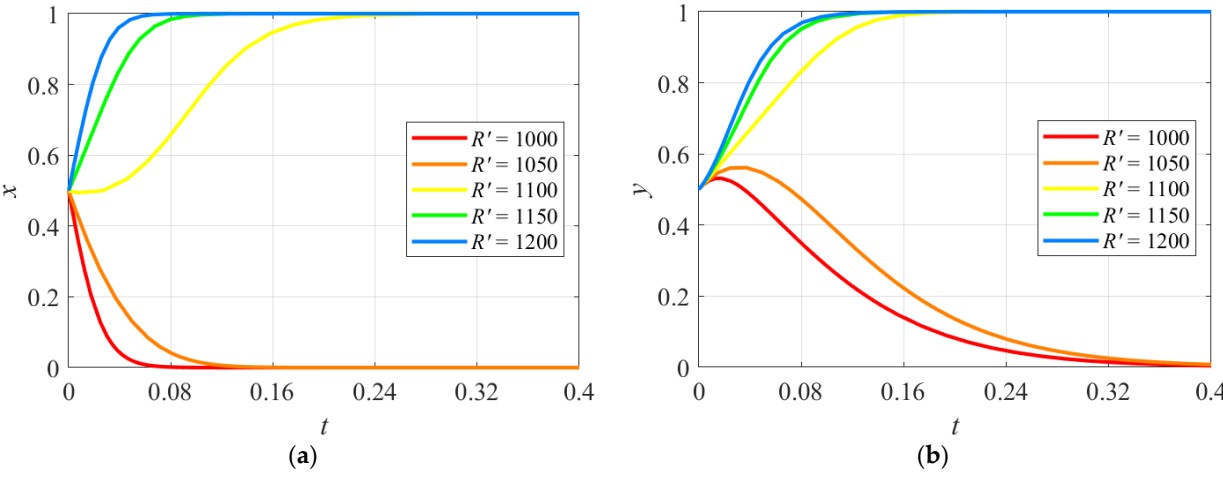

**Figure 5.** Impact of changes in $R'$ on $x$ and $y$. (**a**) Impact of changes in $R'$ on $x$. (**b**) Impact of changes in $R'$ on $y$.

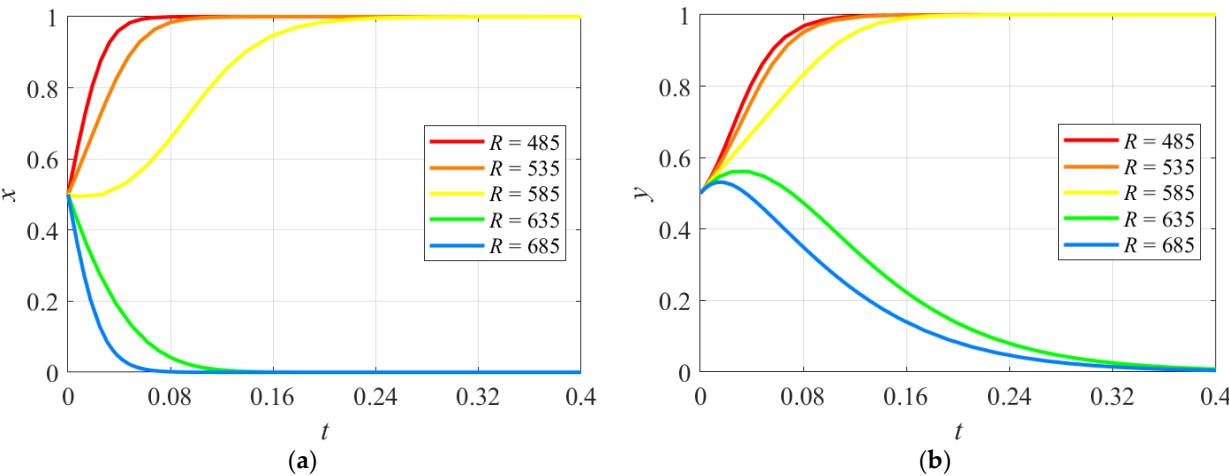

**Figure 6.** Impact of changes in $R$ on $x$ and $y$. (**a**) Impact of changes in $R$ on $x$. (**b**) Impact of changes in $R$ on $y$.

$R'$ and $R$ denote the benefits for original land users, representing the expected return of implementing land redevelopment and the return of maintaining the status quo, respectively. A larger difference between $R'$ and $R$ indicates a greater willingness of original land users to implement industrial land redevelopment, requiring the benefits of redevelopment to surpass the current benefits significantly.

Figure 7 illustrates the substantial impact of changes in demolition and reconstruction costs on $x$ and $y$. As $C_E$ increases, $x$ and $y$ change from 1 to 0, leading to a shift in the selection strategy from (1, 1) to (0, 0). This suggests that heightened demolition and reconstruction costs hinder the implementation of industrial land redevelopment, posing obstacles for original land users when deciding to "implement".

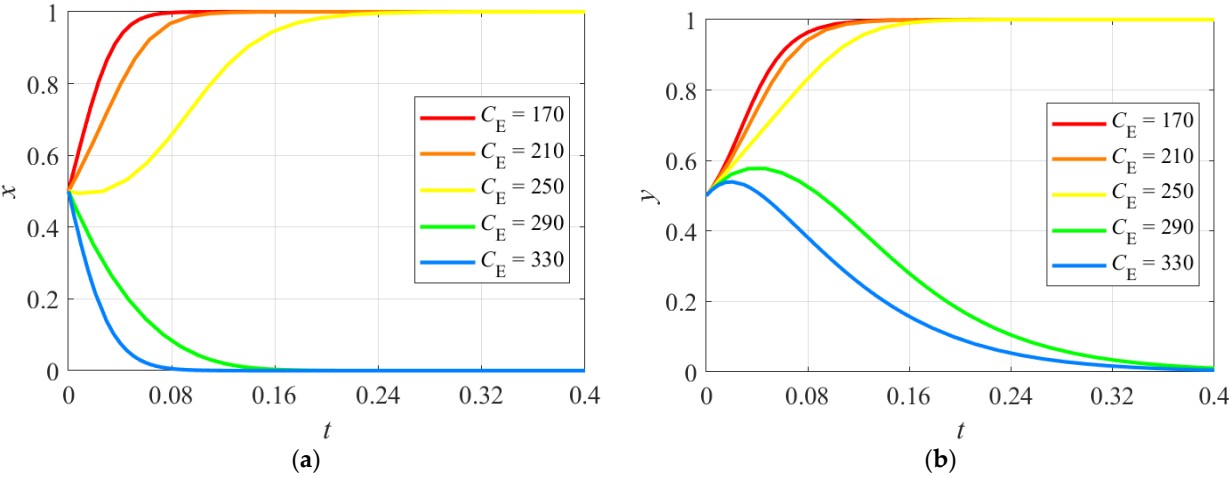

**Figure 7.** Impact of changes in $C_E$ on $x$ and $y$. (**a**) Impact of changes in $C_E$ on $x$. (**b**) Impact of changes in $C_E$ on $y$.

The numerical simulation results above demonstrate the substantial influence of the benefits of maintaining the status quo for original land users, alongside the benefits and costs associated with inefficient industrial land redevelopment, on the evolutionary stability strategy within this study. Lowering implementation costs and enhancing profits for original land users serve as crucial considerations for governmental policy formulation aimed at promoting land redevelopment projects.

4.3.2. Economic Subsidies and Penalties

The effects of changes in parameters $\beta$ and $F$ on $x$ and $y$ are depicted in Figures 8 and 9. While $\beta$ represents the proportion of value-added income of land shared with governments as economic subsidies, $F$ denotes the fines incurred by original land users for refusing to implement land redevelopment when the government actively promotes it. Regardless of the values of $\beta$ and $F$, the final strategy remains (1, 1) when the initial strategy is (0.5, 0.5). However, the time required to reach stability decreases with increasing values of $\beta$ and $F$, indicating a higher likelihood of game players choosing (implement, promote). Hence, we deduce that local government intervention, specifically through the implementation of economic subsidies and punitive measures, facilitates the advancement of inefficient industrial land redevelopment. The effectiveness of incentives and punishments needs further analysis.

This study highlights that original land users exhibit greater sensitivity to parameter changes in comparison to local governments. We investigate the influence of incentives and punishments on the proportion of original land users opting for implementation. When $\beta = 0.16$, the time needed to attain equilibrium is 0.4; when $\beta = 0.24$, the time to reach equilibrium decreases to 0.2. It is evident that a 50% increase in $\beta$ results in a 50% reduction in time. Similarly, when $F$ is 6, the time required for equilibrium is 0.3; when $F$ is 12, the time for equilibrium decreases to 0.25. This indicates that a 100% increase in $F$ leads to a 16.7%

reduction in time. The research findings suggest that the proportion of implementation by original land users is more sensitive to changes in $\beta$. Consequently, we deduce that government economic incentives exert a more significant influence on the strategic decisions of original land users when compared to punitive measures. Both economic subsidies and fines prove advantageous in encouraging industrial land redevelopment by original land users, with economic subsidies playing a more prominent role.

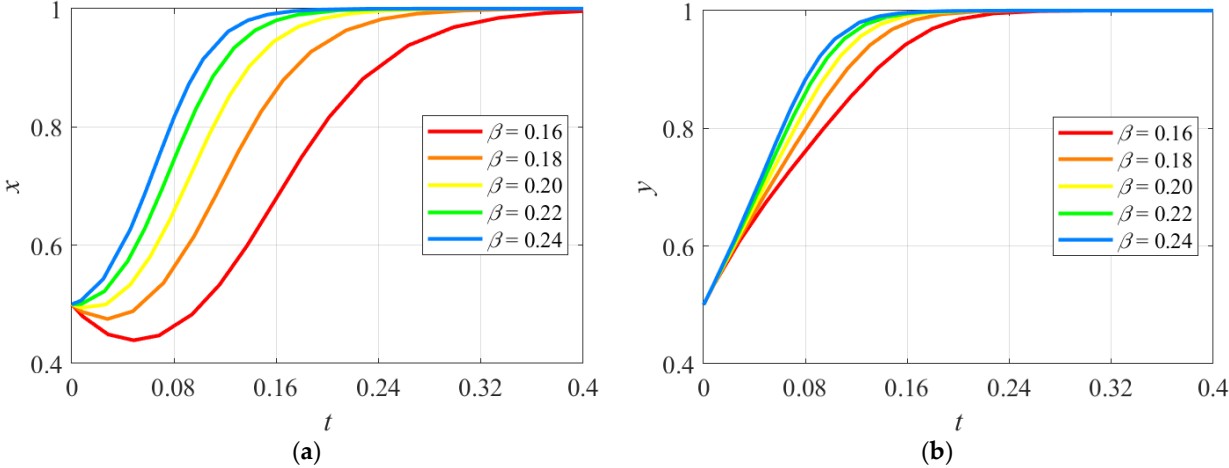

**Figure 8.** Impact of changes in $\beta$ on $x$ and $y$. (**a**) Impact of changes in $\beta$ on $x$. (**b**) Impact of changes in $\beta$ on $y$.

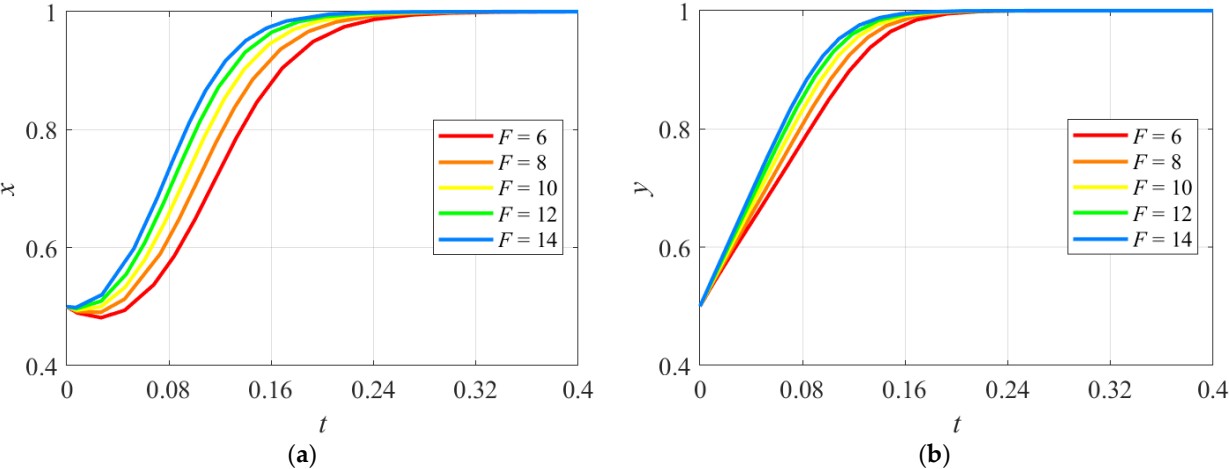

**Figure 9.** Impact of changes in $F$ on $x$ and $y$. (**a**) Impact of changes in $F$ on $x$. (**b**) Impact of changes in $F$ on $y$.

## 5. Conclusions and Policy Implications

This article utilizes evolutionary game theory to establish a game model focusing on the core stakeholders—local governments and original land users—in the urban industrial land redevelopment process, exploring strategies of balancing the interests of these stakeholders. Our study provides several conclusions and policy implications.

Firstly, our research identifies two evolutionary stability strategies within the original land user-led redevelopment model. One strategy involves the original land user implementing industrial land redevelopment, met with a positive response from local governments. Another scenario involves original land users refusing to undertake redevelopment, while local governments abstain from promoting this process. The findings demonstrate that original land users exhibit a deficiency in internal motivation to independently engage in the redevelopment of urban industrial land when solely relying on

market mechanisms. Consequently, we advocate for the proactive involvement of local governments in industrial land redevelopment initiatives, which facilitates a deeper understanding of the demands of original land users, thereby fostering more effective urban renewal processes.

Secondly, through numerical simulation, we further elucidate the influence of initial strategies on game strategies. The results indicate that increased government support positively correlates with the likelihood of industrial land redevelopment implementation by original land users. Our research findings underscore the significant influence of governmental stance on land redevelopment, particularly concerning industrial land. Hence, local governments should adopt a positive and proactive attitude towards the redevelopment of industrial land under the dominant mode of original land users.

Finally, the research reveals that parameter variations significantly impact final stability strategies, with original land user benefits and costs emerging as crucial determinants. Priority is placed on improving the benefits of original land users and reducing related costs as crucial measures to foster industrial land redevelopment. Notably, economic subsidies play a crucial role in the redevelopment of industrial land. While punitive measures cannot change stable strategies, they are effective in increasing the likelihood of original land users choosing redevelopment. This underscores the importance of fiscal incentives and regulatory policies in fostering industrial land redevelopment. Therefore, we recommend the following course of action.

The government should implement measures to lower the expenses incurred by original land users involved in industrial land redevelopment. This entails simplifying the decision-making process and shortening the approval cycle, thus reducing the cost of using funds for the original land users. And, the establishment of a dedicated fund for urban industrial land redevelopment, which provides low-interest loans to projects in compliance with regulations, can alleviate the financial burdens on original land users. Moreover, establishing a benefit-sharing mechanism allowing original land users to partake in land appreciation benefits equitably is essential. While economic subsidies are influential, promoting industrial land renewal cannot rely solely on government financial incentives. The study advocates for the implementation of mandatory systems by the government to incentivize the active participation of original land users in the redevelopment of underutilized industrial land.

This study innovatively analyzes the interest game among core stakeholders in urban industrial land redevelopment decision making under the dominant mode of original land users and explores the key role of government intervention in industrial land redevelopment, enriching land planning theory and industrial land redevelopment practice. Nonetheless, our research has limitations. The decision-making process for industrial land redevelopment involves multiple stakeholders, yet this study only considers local governments and original land users. Future research should explore constructing multi-party game models incorporating additional stakeholders, offering valuable insights into sustainable urban renewal.

**Supplementary Materials:** The following supporting information can be downloaded at: https://www.mdpi.com/article/10.3390/land13040548/s1. Supplementary File: Code of Evolutionary Game.

**Author Contributions:** Conceptualization, F.H. and Y.S.; methodology, Y.S. and Y.Y.; software, Y.S.; validation, Y.Y.; formal analysis, F.H.; investigation, Y.S.; resources, F.H.; data curation, Y.S.; writing—original draft preparation, Y.S.; writing—review and editing, F.H. and Y.Y.; visualization, Y.S.; supervision, F.H. All authors have read and agreed to the published version of the manuscript.

**Funding:** This research received no external funding.

**Data Availability Statement:** The data presented in this study are available on request from the corresponding author. The data are not publicly available due to privacy restrictions.

**Conflicts of Interest:** The authors declare no conflicts of interest.

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
