# Peer review of "Evolution Process of Urban Industrial Land Redevelopment in China: A Perspective of Original Land Users"

_land, doi:10.3390/land13040548_

Round 1

Reviewer 1 Report

Comments and Suggestions for Authors

This article proposes an evolutionary game model to investigate the interest game between local governments and land owners in the context of urban industrial land redevelopment. The article presents a study of the different equilibrium points and of the stability of the strategies for different scenarios. Finally, the article provides and empirical validation using numerical simulations before deriving implications of the model for policy making.

The article is interesting, quite clear, well written and illustrated.                       

I am quite confused about the term "original land users". It seams to me the authors are actually referring to land owners. Did I miss something?

I would have appreciated a discussion on the potential validation of the model with real world data: would it be possible to validate such a model using known policies?

Furthermore, I would be interested in the generalisation of the proposed approach to other contexts: could the model be adapted to other contexts than China? Would that make sense?

In terms of open science, I suggest that the code used for numerical simulations might be released as open source code.                                           

In my opinion, tables 2 & 8 are not clear enough: either state the payoff (olu/gov) for each cell or make 2 tables.                                                                  

Author Response

Thank you very much for your valuable suggestions. Please see the attachment.

Reviewer 2 Report

Comments and Suggestions for Authors

Aim of the work is to utilize evolutionary game theory to investigate the interest game between local governments and original land users in the model of urban industrial land redevelopment dominated by original land users. Some comments are here provided in order to improve the work:

Abstract

-summarize the results and higlight the innovative contributions to the reference field

Author Response

(The authors gave the same response as above.)

Reviewer 3 Report

Comments and Suggestions for Authors

The authors apply game theory to model the relationship between local governments and land users to drive urban industrial land redevelopment dominated by original land users. It explores the impact of incentives and mandatory regulations on industrial land redevelopment. This is an important and interesting topic of sustainable land development that can provide insight into policies and decision making regarding future land development for the urban social and economic progress. For this purpose, strategic planning is important and it is useful to have tools that quantify the influence of different factors.

The paper is well written and appropriately structured. Methods that are used are clearly presented as well as obtained results.

The authors should explain in more detail why evolutionary game theory was chosen and what are the advantages over other approaches. It would be good to summarize main findings as a set of recommendations. The main findings are somewhat general emphasizing pivotal government role in driving land development change. It would be also good to provide some quantitative results in the abstract or conclusion supporting these findings.

The authors mention that results have been empirically validated using Matlab. Is it possible to automate this approach in a form a software tool to help decision makers make appropriate decision?

Author Response

(The authors gave the same response as above.)

Reviewer 4 Report

Comments and Suggestions for Authors

Authors present potentially very interesting research about the urban industrial land redevelopment role in the achievement of a sustainable development pattern in urban renewal strategies and actions. In particular, the paper investigates the function of governments and original land users, aiming to provide a scientific basis for policy formulation to achieve a balanced alignment of interests between the two parties. Authors underline the missing research in the current national (in China) and international debate on the redevelopment of industrial areas: the impact of government interventions such as economic incentives and punitive measures. This represents an important aspect to better analyze and assess the process of industrial sites renewal (in a sustainable way): time, stakeholders involved, new urban functions, policies definitions, economic impacts for the contractor, social impact for the neighborhood, environmental benefits and so on. The conclusions remark the evidence obtained. The main topic fits the Journal scope.
The major issues detected are:
1) please revise the references list (at the moment not satisfactory), adding more recent and international authors and research on the main fields as industrial/brownfields renewal, sustainable development, SDGs goals and so on;
2) the introduction section is focused on the Chinese situation mostly. Moreover, in my opinion, the literature review section has not a clear focus (in particular sub-sections 2.1 and 2.3). Please try to align the sections and to better specify the case study environment (from geographical, social, economic and urban planning point of view);
3) regarding the methodology: in chapter 3, I strongly suggest adding a logical framework to underline the methodological steps of your research. After this, explain the different used instruments to reach your goal;
4) regarding the methodology: in my opinion it is better to summarize sub-sections 3.2, 3.3 and 3.4 and better describe and analyze the real application on your case study (chapter 4).
5) I strongly suggest to not use Figures 1-4 (not necessary) and to focus on the quality (increasing the dimensions and the clarity of each element in the charts) of Figures 5-12 (that represent the real core of the methodology’s application).

Comments on the Quality of English Language

I recommend a proofreading to make the sentences simpler: they are often difficult to read due to the complexity of their structure.

Author Response

(The authors gave the same response as above.)

Round 2

Reviewer 4 Report

Comments and Suggestions for Authors

In my opinion, authors provide a new satisfactory verison of their paper.

Just a comment: it could be easier to have also a graphic representation of the methodological framework.